# Tuning excited state electronic structure and charge transport in covalent organic frameworks for enhanced photocatalytic performance

Zhongshan Chen[1], Jingyi Wang[1], Mengjie Hao[1], Yinghui Xie[1], Xiaolu Liu[1], Hui Yang [1]✉, Geoffrey I. N. Waterhouse [2], Xiangke Wang [1]✉ & Shengqian Ma [3]✉

Covalent organic frameworks (COFs) represent an emerging class of organic photocatalysts. However, their complicated structures lead to indeterminacy about photocatalytic active sites and reaction mechanisms. Herein, we use reticular chemistry to construct a family of isoreticular crystalline hydrazide-based COF photocatalysts, with the optoelectronic properties and local pore characteristics of the COFs modulated using different linkers. The excited state electronic distribution and transport pathways in the COFs are probed using a host of experimental methods and theoretical calculations at a molecular level. One of our developed COFs (denoted as COF-4) exhibits a remarkable excited state electron utilization efficiency and charge transfer properties, achieving a record-high photocatalytic uranium extraction performance of ~6.84 mg/g/day in natural seawater among all techniques reported so far. This study brings a new understanding about the operation of COF-based photocatalysts, guiding the design of improved COF photocatalysts for many applications.

Covalent organic frameworks (COFs), owing to the programmability and tunability of their composition, structure and porosity, are finding increasing application in adsorption[1], sensing[2], catalysis[3–5], energy storage/conversion[6,7], and environmental remediation[8]. COFs demonstrate excellent potential for photocatalytic applications owing to the vast array of linkers available as light absorption components, electron donors, electron acceptors, etc[9–13]. However, fast charge (electron-hole) recombination, instability of transient species, and energy losses during charge excitation and migration can greatly reduce the photocatalytic activity of COFs, with the efficient separation of photogenerated charges requiring the use of sacrificial reagents. These drawbacks hinder the practical development of COF-based photocatalysts.

The development of highly active and low-cost COF photocatalysts is now an international research focus, bridging the fields of chemistry, material science, catalysis, and engineering[9–13]. To obtain high-performance COF photocatalysts, researchers are now exploring ways of increasing the visible-light absorption range[14–16], optimizing band structures[17–20], and decreasing the recombination of photogenerated electrons and holes[21–23]. Common strategies for achieving these performance-boosting properties include (i) incorporating a photosensitizer into the framework for improving the light-harvesting capability[24–27]; (ii) functionalization of the linkers and tuning of the components to optimize the band gap energy and valence/conduction band potentials[17,28–31]; (iii) construction of donor-acceptor moieties to

[1]College of Environmental Science and Engineering, North China Electric Power University, Beijing 102206, P. R. China. [2]School of Chemical Sciences, The University of Auckland, Auckland 1142, New Zealand. [3]Department of Chemistry, University of North Texas, Denton, TX 76201, USA. ✉e-mail: h.yang@ncepu.edu.cn; xkwang@ncepu.edu.cn; sqma@usf.edu

improve charge transfer kinetics and charge carrier separation efficiencies[32–35]; (iv) doping non-metal elements (N, P, S, etc.)[36,37], single metal sites[38–41], clusters[41–43], or noble metals[29] as a co-catalyst to modulate the photoelectronic properties, thus improving the overall photocatalytic activity. These approaches demonstrate that controlling electron energy levels and electron transport in COFs is vital to improving photocatalytic performance.

In recent years, the extraction of uranium from seawater has attracted increasing attention since nuclear energy can help curb fossil fuel dependencies and reduce anthropogenic $CO_2$ emissions[44,45]. Discovering sustainable ways of harvesting uranium from seawater is critical to ensuring a reliable supply of uranium fuel for future generations (as well as being of importance in the treatment of wastewater from the nuclear industry and contaminated groundwater). COF-based adsorbent-photocatalyst systems are now being pursued for uranium extraction[46–48]. To date, the developed systems have relatively low activity for U(VI) reduction and/or need sacrificial reagents for separating photo-generated charges. Further, their complicated structures

result in indeterminacy of catalytic sites, hindering efforts to understand reaction mechanisms that would enable rational COF photocatalyst design. A further unmet challenge in this domain is the limited sunlight absorption and charge carrier utilization efficiency, both of which depend on COF structure and electron transport pathways at a molecular level (Fig. 1a, b). Decreasing the energy losses during electron transfer from the excited state to the acceptor (e.g., adsorbed $UO_2^{2+}$) while reducing fluorescence emissions are essential for improving photocatalytic performance (Fig. 1a, b).

Herein, we synthesized a series of isoreticular COFs with different excited state electron distributions, charge transport properties and local pore characteristics (Fig. 1c), which we subsequently evaluated for the photocatalytic reduction of U(VI) to U(IV) solid (such as $UO_2$) from seawater and contaminated groundwater. COF-3 and COF-4 with high-symmetry $C_3$-linkers exhibited efficient photocatalytic activities for aqueous uranyl reduction to $UO_2$. COF-1 and COF-2 containing asymmetric $C_3$ ligands showed relatively low activities. Mechanistic studies showed that COF-1, COF-2, COF-3, and COF-4 differed in their

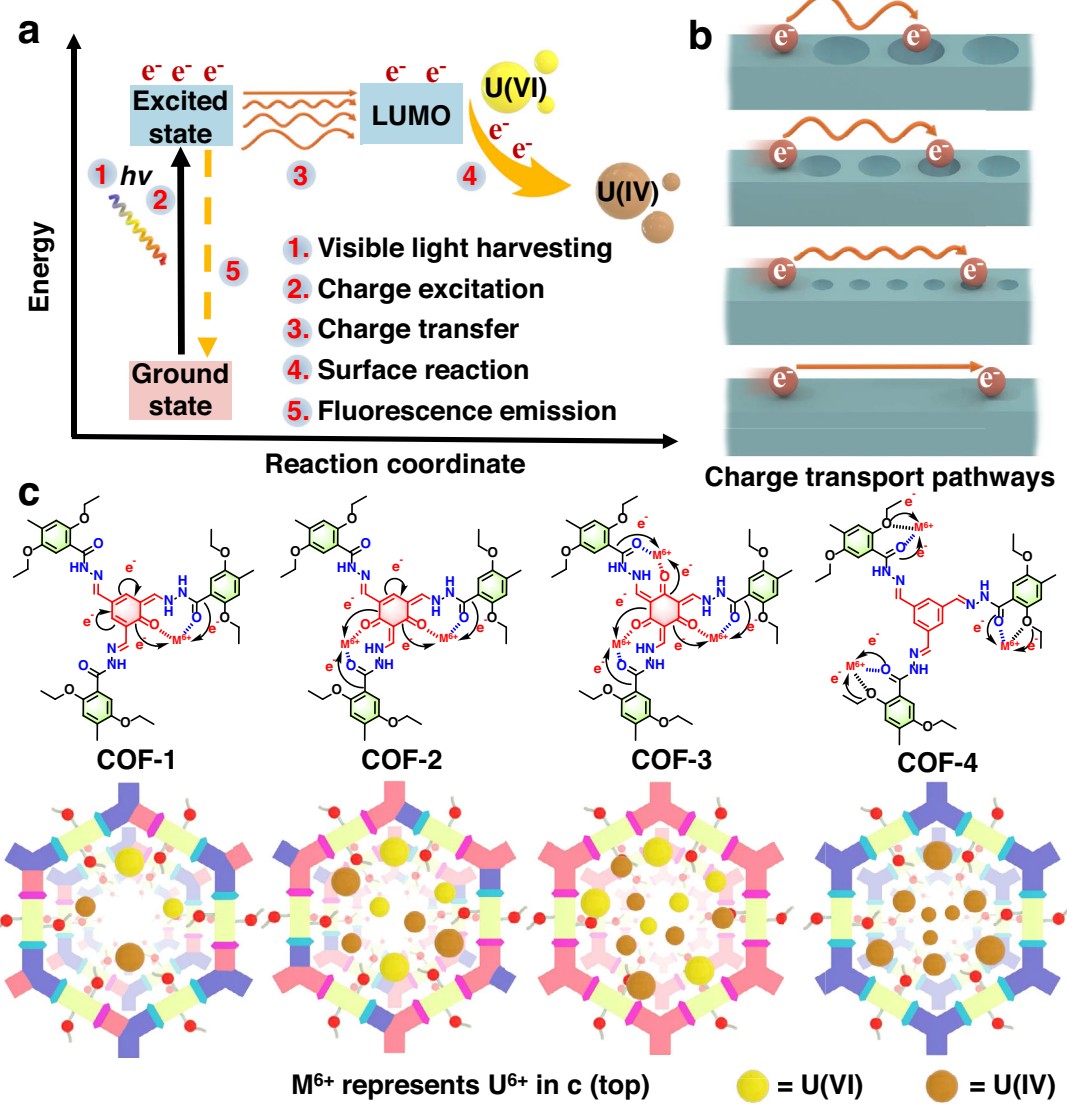

**Fig. 1 | Overview of the electron excitation processes and charge carrier utilization in COFs. a** Schematic illustration of charge carrier separation and utilization by COF photocatalysts, highlighting the energy levels involved in the charge transfer processes. **b** Illustration of charge transfer from a COF donor to a U(VI) acceptor (highlighting the different utilization efficiency), which can be optimized by tuning the excited state electronic distribution. **c** Structures of COF-1 to COF-4 with different components in the pores to modulate the excited state electronic structure and charge transport properties, thus tuning the photocatalytic activities for uranium extraction.

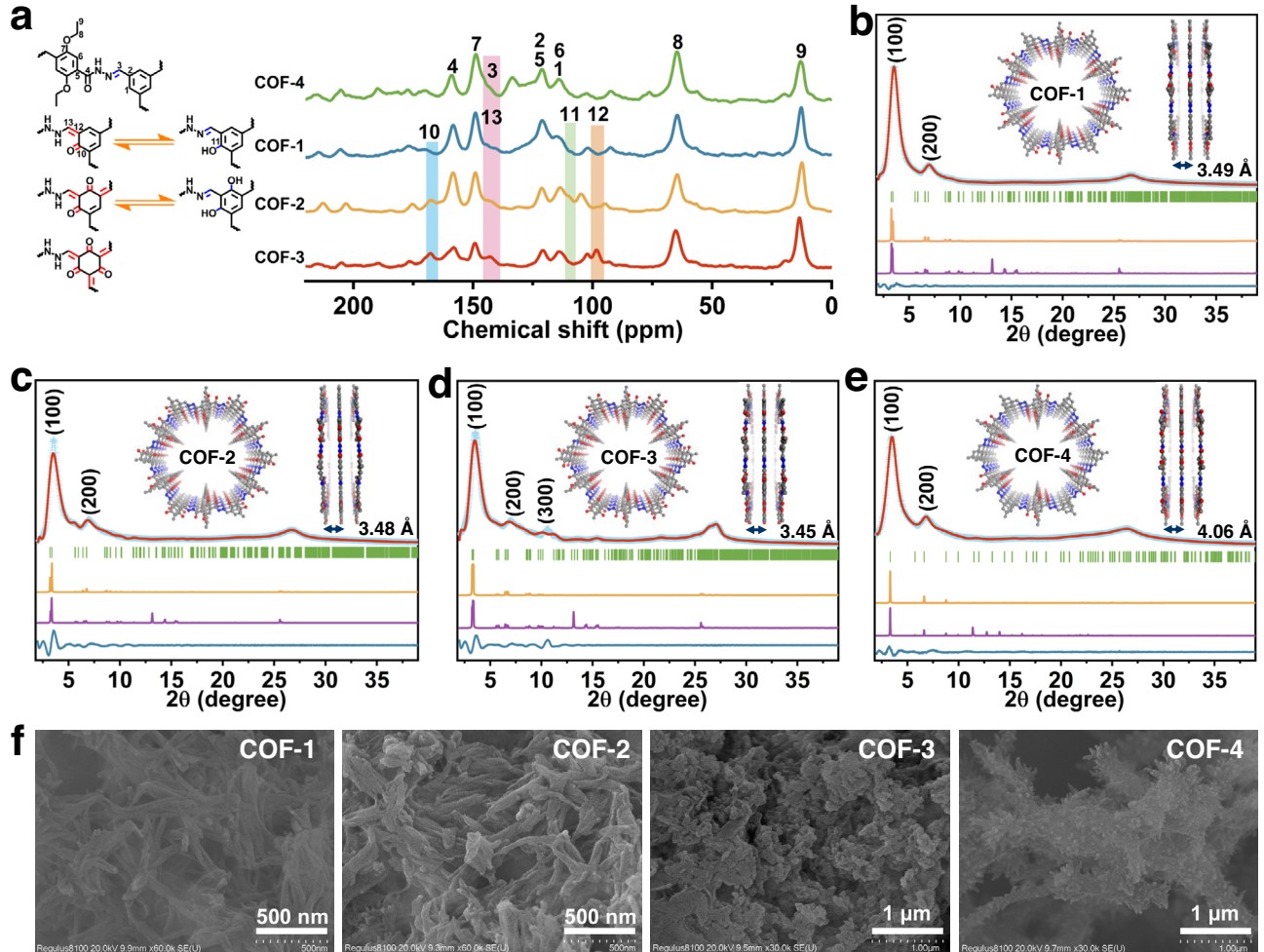

**Fig. 2 | Chemical structure and characterization of COFs. a** Solid-state [13]C CP/ MAS NMR spectra of COF-1, COF-2, COF-3, and COF-4. **b–e** Experimental and simulated PXRD patterns of COF-1, COF-2, COF-3, and COF-4 with corresponding Pawley refinements (red), simulated eclipsed AA stacking results (khaki), simulated staggered AB stacking results (purple), and Bragg positions (green) showing that the AA stacking mode provided a good fit to the experimental data (azure) with minimal differences (dark blue). The insets show the structural models and inter-layer distances of each COF assuming a AA stacking mode. **f** SEM images of COF-1, COF-2, COF-3, and COF-4.

electron distributions, electron donating sites, electron transport pathways, and free energy profiles from the donor to the U(VI) acceptor. Notably, the active sites of COF-4 were beneficial for electron-hole pair separation and efficient charge carrier utilization, resulting in a record-high photocatalytic uranium extraction efficiency in natural seawater. These results demonstrate an effective approach for tuning the excited electronic structure and electron transportation of COF materials to enhance photocatalytic extraction of uranium from both seawater and contaminated groundwater. Further, results offer new insights about COF photocatalytic mechanisms at a molecule level.

## Results

### Design, synthesis, and characterization of COFs

Our synthetic strategy to develop isoreticular COFs with different excited state electronic structures and local pore characteristics was based on the use of different linkers (Fig. 1c). Firstly, we synthesized hexagonal hydrazide-based COFs with varying numbers of $\beta$-ketoe-namine-imine moieties located on the surface of micropores, obtaining COF-1, COF-2, and COF-3, via condensation of 2,5-die-thoxyterephthalohydrazide (DETH) with 2-hydroxybenzene-1,3,5-tri-carbaldehyde (TFP), 2,4-dihydroxybenzene-1,3,5-tricarbaldehyde (DTFP), and 1,3,5-triformylphloroglucinol (TP), respectively, in a

mixture of mesitylene and 1,4-dioxane with acetic acid as the catalyst at 120 °C (see "Methods" section). Next, we reacted 1,3,5-tri-formylbenzene (TFB) and DETH under similar solvothermal condi-tions at 120 °C to obtain COF-4 without any carbonyl groups on the pore walls. The structures of COF-1, COF-2, COF-3, and COF-4 were determined by solid-state cross-polarization magic angle spinning [13]C nuclear magnetic resonance (CP/MAS [13]C-NMR) spectroscopy, Fourier transform infrared (FT-IR) spectroscopy, powder X-ray dif-fraction (PXRD), and structural simulations in Materials Studio. The [13]C-NMR of COF-1, COF-2, COF-3, and COF-4 revealed the formation of C=N – /C=C – (142.6 ppm) and O=C – NH – (158.8 ppm), confirming the successful reaction of the starting reagents (Fig. 2a). FT-IR signals for C=C – NH – /C=N – NH – at -1527 cm⁻¹ were consistent with the [13]C-NMR results (Supplementary Fig. 1). The disappearance of the −CH=O stretches at -1643 cm⁻¹ and −NH₂ signals at 3320 cm⁻¹ further confirmed the conversion of the aldehyde and hydrazine groups in the reactants. The presence of different amounts of ketoenamine moieties (keto tautomers) in COF-1, COF-2, and COF-3 through reversible or irreversible $\beta$-ketoenamines reactions were supported by the disappearance of the −OH signals at 1249 cm⁻¹ and progressive increase of C=O signals at -1623 cm⁻¹, respectively (Supplementary Fig. 1)[49,50]. Moreover, the gradual decrease of C=N stretching peaks at 1670 cm⁻¹ for COF-1 to COF-3 further verified the conversion of enol-

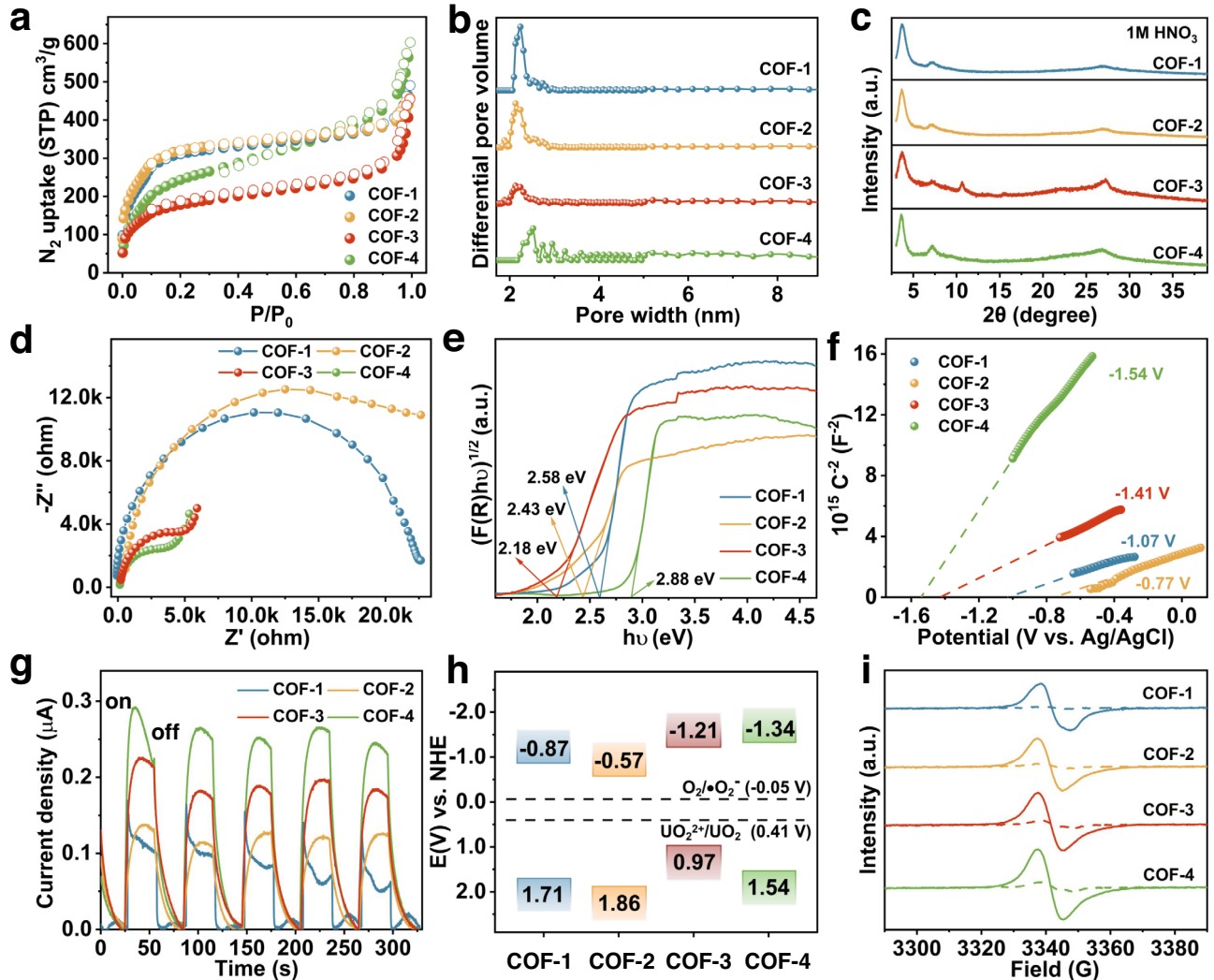

**Fig. 3 | Porosity, stability and optoelectronic properties of COFs. a** N$_2$ sorption isotherms measured at 77 K for COF-1, COF-2, COF-3, and COF-4. **b** Pore size distribution calculated using a DFT method from N$_2$ isotherms measured at 77 K for COF-1, COF-2, COF-3, and COF-4. **c** PXRD patterns of COF-1, COF-2, COF-3, and COF-4 after treatments in 1 M HNO$_3$ over 24 h. **d** Electrochemical impedance spectra (EIS) of COF-1, COF-2, COF-3, and COF-4. **e** Kubelka-Munk plots of COF-1, COF-2, COF-3, and COF-4. **f** Mott-Schottky plots of COF-1, COF-2, COF-3, and COF-4. **g** Transient current density of COF-1, COF-2, COF-3, and COF-4. **h** Energy band position of COF-1, COF-2, COF-3, and COF-4. **i** EPR conduction band electron spectra of COF-1, COF-2, COF-3, and COF-4 in the dark and under visible light irradiation.

imine (OH) to keto-enamine (C=O) via reversible or irreversible proton tautomerism. $^{13}$C-NMR spectra showed C=O peaks around 168.1 ppm in the spectra of COF-1, COF-2, and COF-3, providing strong evidence for newly formed ketone moieties (Fig. 2a). The signals at -98.5 ppm intensified on going from COF-1 to COF-3, further suggesting the presence reversible and irreversible ketoenamine-imine moieties. The peaks at -12.9 ppm and -64.5 ppm are assigned to ethoxy groups from the DETH linker.

Next, we determined the crystal structures of the four COFs using PXRD and structural simulations. As shown in Fig. 2b, COF-1 showed peaks at 3.5° and 6.9° could be assigned to the (100) and (200) facets, respectively. Structural simulation results by Materials Studio[51] further suggested that it crystallized in space group $P1$ with 2D eclipsed (AA) stacking, with the Pawley refinement showing a negligible difference between the simulated and experimental PXRD patterns (Fig. 2b and Supplementary Table 1). The unit cell parameters of $a = 30.30$ Å, $b = 30.54$ Å, $c = 3.49$ Å, $\alpha = \beta = 90°$, $\gamma = 117.98°$ with negligible residuals $R_{wp} = 2.72\%$ and $R_p = 1.58\%$. On the basis of these results, it was determined that COF-1 possessed a 2D structure with honeycomb-like pores (Fig. 2b and Supplementary Fig. 2). The

interlayer distance was calculated to be approximately 3.49 Å. COF-2 had the same space group, but slightly different cell parameters (Fig. 2c, Supplementary Fig. 3, and Supplementary Table 2; $a = 30.10$ Å, $b = 31.23$ Å, $c = 3.48$ Å, $\alpha = \beta = 90°$, $\gamma = 119.33°$, residuals $R_{wp} = 7.21\%$, and $R_p = 4.54\%$). COF-3 was assigned to the $P3$ space group with optimized parameters of $a = b = 30.51$ Å, $c = 3.92$ Å, $\alpha = \beta = 90°$, $\gamma = 120°$, residuals $R_{wp} = 6.22\%$ and $R_p = 3.96\%$ from the AA stacking model (Fig. 2d, Supplementary Fig. 4, and Supplementary Table 3). COF-4 possessed a similar geometrical structure with optimized parameters of $a = b = 30.81$ Å, $c = 4.06$ Å, $\alpha = \beta = 90°$, $\gamma = 120°$, with $R_{wp} = 2.27\%$ and $R_p = 2.15\%$ (Fig. 2e, Supplementary Fig. 5, and Supplementary Table 4). These results revealed that COF-1, COF-2, COF-3, and COF-4 were an isoreticular family of well-ordered frameworks. However, it is interesting to note that the hexagonal channels of these COFs are parallel to each other in 3D frameworks with different local pore characteristics based on different C$_3$ linkers (Supplementary Figs. 2–5). The COFs exhibited good thermal stability, with thermal decomposition temperatures above 350 °C under an N$_2$ atmosphere (Supplementary Fig. 6). Scanning electron microscopy (SEM) images showed a mixture of nanofiber and nanoparticle

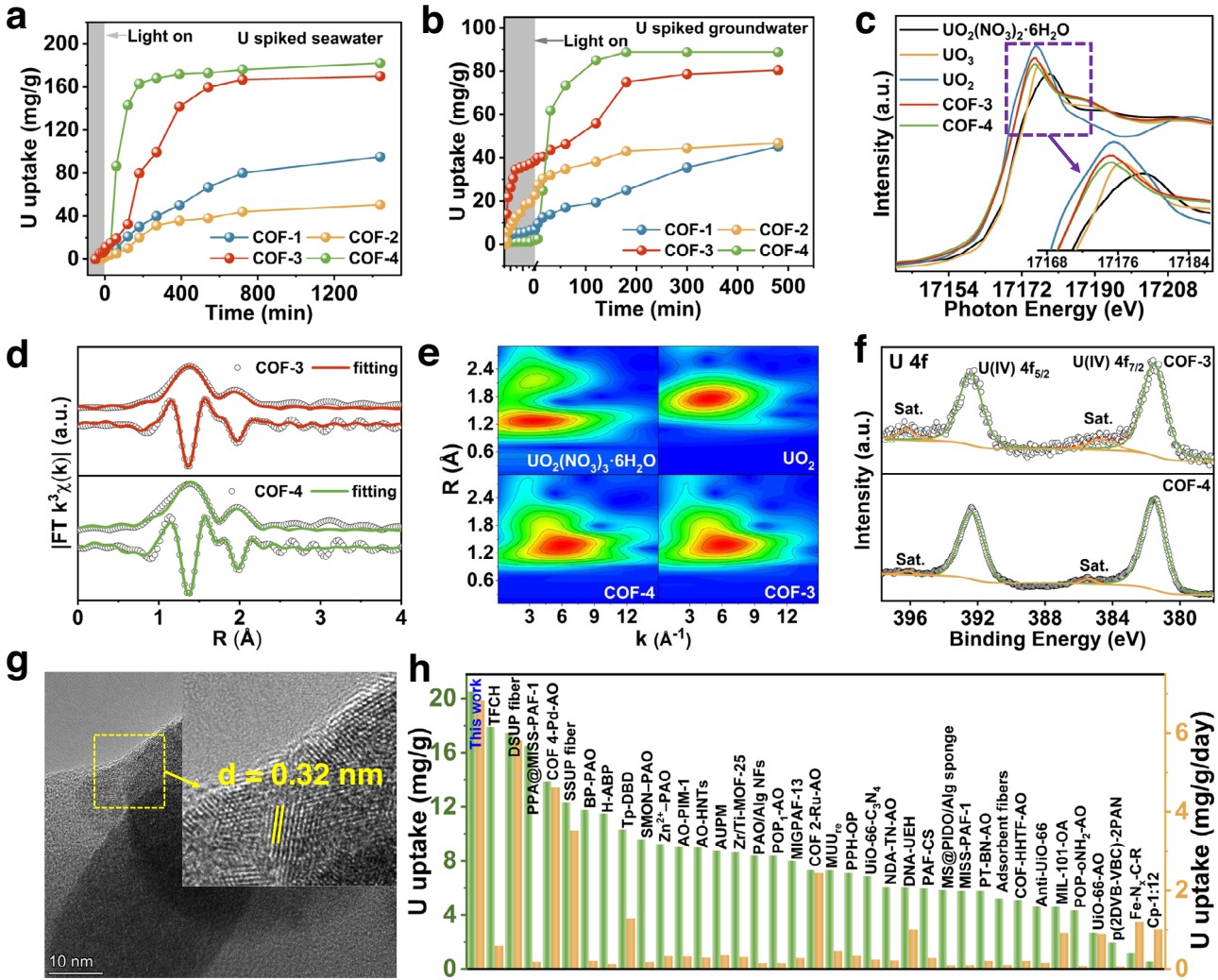

**Fig. 4 | Photocatalytic uranium extraction performance. a** Uranium extraction from spiked seawater with initial uranium concentrations of ~20 ppm, using COF-1, COF-2, COF-3, and COF-4 as photocatalysts. **b** Uranium extraction from spiked groundwater with initial uranium concentrations of ~10 ppm, using COF-1, COF-2, COF-3, and COF-4 as photocatalysts. **c** U L$_{III}$-edge XANES spectra for COF-3 and COF-4 after uranium extraction studies. UO$_2$(NO$_3$)$_2$·6H$_2$O, UO$_2$, and UO$_3$ are employed for comparison. **d** EXAFS fitting curve for COF-3 and COF-4 after

photocatalysis. **e** WT contour plots for COF-3 and COF-4. UO$_2$(NO$_3$)$_2$·6H$_2$O and UO$_2$ are employed for comparison. **f** U 4$f$ XPS spectra of COF-3 and COF-4 after photocatalysis. **g** HRTEM image of COF-4 (attached a solid nanoparticle) after photocatalysis in uranium-spiked seawater. **h** Comparison of uranium extraction performance of COF-4 and other reported materials in natural seawater. The reference data for UO$_2$(NO$_3$)$_2$·6H$_2$O, UO$_2$, and UO$_3$ in **c** and **e** were taken from our previous work[47,70].

morphologies for COF-1, COF-2, COF-3, and COF-4, respectively (Fig. 2f).

## Porosity and chemical stability of COFs

The COFs reported herein were generated using the principles of reticular chemistry, with the local pore characteristics able to be systematically tuned by the functional groups in the linkers. The porosity of each COF was evaluated by conducting nitrogen sorption isotherms on fully activated samples at 77 K. The adsorption-desorption isotherms were classified as type-II curves with mesoporous characteristics (Figs. 3a and 3b). The calculated Brunauer-Emmett-Teller (BET) surface areas were 1063.6, 1084.9, 616.8, and 878.9 m²/g for COF-1, COF-2, COF-3, and COF-4, respectively. The total pore volumes were estimated to be 0.56, 0.57, 0.41, and 0.64 cm³/g for COF-1, COF-2, COF-3, and COF-4, respectively. The calculated pore sizes were 2.24, 2.17, 2.09, and 2.51 nm, for COF-1, COF-2, COF-3 and COF-4, respectively, in good general agreement with the observed pore sizes determined in the crystal structure simulations (Fig. 3b). We further examined the chemical stability of all the COFs by immersing them in natural

seawater and acidic solutions. All COFs demonstrated excellent chemical stability in natural seawater or 1 M HNO$_3$ over 24 h, evidenced from PXRD and FT-IR results (Fig. 3c and Supplementary Figs. 7 and 8).

## Optoelectronic properties

Electrochemical impedance spectroscopy (EIS), ultraviolet/visible (UV-Vis) diffuse reflectance spectroscopy, Mott-Schottky plots, photocurrent curves, and electron paramagnetic resonance (EPR) spectroscopy analyses were conducted to evaluate the optoelectronic properties of the obtained COFs. The EIS-Nyquist plots showed COF-3 and COF-4 to have the smallest semicircle diameter, indicating those two COFs possessed superior electrical conductivity to COF-1 and COF-2 (Fig. 3d). As shown in Supplementary Fig. 9, COF-1, COF-2, COF-3, and COF-4 absorbed strongly at visible wavelengths, with the photoabsorption edges at 518, 580, 481, and 430 nm, respectively. The corresponding bandgaps of COF-1, COF-2, COF-3, and COF-4 were calculated to be 2.58, 2.43, 2.18, and 2.88 eV, respectively, using the Kubelka-Munk function, suggesting their robust visible-light harvesting (Fig. 3e). The Mott-Schottky plots showed the conduction bands

(CB) levels for COF-1, COF-2, COF-3, and COF-4 to be −1.07, −0.77, −1.41, and −1.54 V (vs. Ag/AgCl), respectively (Fig. 3f). Subsequently, photocurrent measurements were performed on the COFs to assess the separation efficiency of the photogenerated electron-hole pairs. COF-3 and COF-4 showed the strongest photocurrent responses (Fig. 3g). Based on these characterization results, we summarized the energy band position of these COFs, among which the valence bands (VB, vs. NHE) of COF-1, COF-2, COF-3, and COF-4 were at 1.71, 1.86, 0.97, and 1.54 V, respectively (Fig. 3h)[52]. Importantly, the CB level of all four COFs was more negative than the U(VI)/U(IV) redox couple (0.41 V vs. NHE) (Fig. 3h), enabling the photoreduction of $UO_2^{2+}$ to a $UO_2$ solid under bandgap excitation. EPR spectroscopy was used to verify the presence of unpaired electrons in the conduction band of the COFs under visible light excitation. Compared to the dark condition, all four COFs exhibited signals with $g = 2.004$ upon visible-light excitation, indicating the presence of unpaired electrons in the conduction band (Fig. 3i). Taken together, these results showed the COFs possessed good light harvesting ability, electron conductivity, charge separation and transport properties, particularly COF-4. The difference in these properties between the COFs could be attributed to the local pore characteristics decorated with varying components attached to the photoactive units. This suggested that the electron carrier utilization efficiency in the COF photocatalysts could be systematically tuned by controlling and regulating the local pore characteristics, thus enhancing their photocatalytic performance. To explore the practical potential of our approach, we next carried out a series of experiments to assess the photocatalytic performance of the developed COF photocatalysts towards uranium extraction.

## Photocatalytic uranium extraction studies

Uranium extraction from seawater and contaminated groundwater is of prime importance for fuel supply and environmental remediation, yet technically is very challenging. Physicochemical adsorption methods show promise for uranium extraction from seawater and wastewater[53–63]. However, the developed sorbents have limitations, including relatively poor uranium selectivity relative to other metal ions, slow kinetics, poor durability, serious biofouling, high cost, and/or harsh conditions elution for adsorbent regeneration. In this context, we performed detailed photocatalytic uranium extraction studies on the aforementioned COFs in spiked seawater, groundwater, and natural seawater to evaluate their performance for uranium extraction. No sacrificial electron donor reagents were used in the reaction system. Initial experiments were conducted in ~20 ppm uranium-spiked seawater at 25 °C (the pH was adjusted to ~8 by adding $Na_2CO_3$). Before turning on the Xenon lamp, the COFs were immersed in the uranium-spiked seawater overnight to achieve adsorption equilibrium. The uranium is mainly in the speciation of $[UO_2(CO_3)_3]^{4-}$ in seawater, and the adsorbent needs to compete with carbonate groups for binding uranyl ions[55,57,60,61,63,64]. In comparison with the binding affinity of $CO_3^{2-}$ to $UO_2^{2+}$, hydrazine-carbonyl groups through COFs 1 to 3 and the hydrazine sites in COF-4 showed stronger binding affinities towards $UO_2^{2+}$ than towards $CO_3^{2-}$, therefore the $[U(VI)O_2(H_2O)_2]^{2+}$ adsorbed spontaneously on all the COFs[64].

The collected photocatalytic data is summarized in Fig. 4a. COF-4 quickly removed uranium from spiked seawater, with an uptake capacity of 182 mg/g U achieved over 24 h. COF-3 was slightly less efficient than COF-4, affording a capacity of 170 mg/g U (24 h). COF-1 and COF-2 showed much lower activities under similar conditions. The photocatalytic activity of these COFs was further evaluated by conducting uranium recovery experiments in uranium-spiked groundwater (~10 ppm), with the catalytic efficiency also following the order COF-4 (capacity up to 89 mg/g in 3 h) > COF-3 > COF-1 > COF-2 (Fig. 4b and Supplementary Fig. 10). Notably, COF-4 exhibited the lowest adsorption capacity but the highest photocatalytic activity among all of these COFs, which suggested very efficient charge carrier

generation and utilization under visible light irradiation (Supplementary Fig. 10). To characterize the uranium-containing products of the photocatalytic experiments, U $L_{III}$-edge X-ray absorption near edge spectroscopy (XANES), Fourier-transformed extended X-ray absorption fine structure spectroscopy (FT-EXAFS), wavelet transform (WT) contour plots, X-ray photoelectron spectroscopy (XPS), and high-resolution transmission electron microscopy (HRTEM) measurements were performed on COF-3 and COF-4 after photocatalysis. The U $L_{III}$-edge XANES spectra for the used photocatalysts were similar to that of the $UO_2$ reference sample (Fig. 4c), revealing that the photocatalytic reaction generated a U(IV) product. The FT-EXAFS spectra of both used COFs exhibited peaks at ~1.42 Å and 1.97 Å, which could be readily assigned to the U-O bonding (Fig. 4d and Supplementary Table 5). The fitted curves indicated that the U(IV) coordination environment in the product was consistent with the formation of $UO_2$ (Fig. 4d and Supplementary Table 5). The WT contour plots of COF-3 and COF-4 after photocatalysis showed a WT maximum in k space of 6.2 Å$^{-1}$ and R space at 1.34 Å, closely matching data for the $UO_2$ reference powder (Fig. 4e). U $4f$ XPS spectra revealed U $4f_{7/2}$ and U $4f_{5/2}$ peaks at 381.6 and 392.3 eV, respectively, corresponding to a U(IV) species (Fig. 4f). The HRTEM image of COF-4 after photocatalysis showed small nanoparticles with lattice fringe spacing of 0.32 nm, corresponding to the (111) plane in a cubic $UO_2$ (Fig. 4g and Supplementary Fig. 11). Results conclusively demonstrate that U(VI) in both spiked seawater and groundwater could be photocatalytically reduced to $U(IV)O_2$ by the developed COFs under visible light irradiation, with the latter being easily collected by mechanical agitation (sonication) of the COFs. In addition, COF-4 showed outstanding cycling photochemical durability and activity in seawater with minimal activity decay after six cycles of photocatalysis experiments (Supplementary Fig. 12). FT-IR and PXRD results revealed the initial structure of frameworks were retained after reuse, confirming the high photochemical stability of COF-4 (Supplementary Figs. 13 and 14). The $N_2$ adsorption-desorption isotherms and calculated BET surface area of 547.4 m²/g confirmed the good stability of COF-4 after photocatalysis (Supplementary Fig. 15).

Next, we conducted detailed studies of uranium extraction by COF-4 in natural seawater. As expected, COF-4 exhibited rapid uranium removal performance in natural seawater with an uptake capacity as high as 20.6 mg/g after 72 h (Fig. 4h). Next, the selectivity of uranium uptake was explored. Copper and vanadium ions are known to be particularly problematic to the development of adsorbents for uranium (including the state-of-art amidoxime-based materials). Therefore, we evaluated the selectivity of COF-4 photocatalysis for uranium extraction from natural seawater in the presence of different ions. Impressively, COF-4 showed a high selectivity for uranyl ions over other cations including copper, vanadium, zirconium, iron, nickel, lead, and cobalt ions (Supplementary Fig. 16). Next, we compared the uranium extraction performance of the COF-4 with other recently reported COF materials (Fig. 4h and Supplementary Table 6). The uranium extraction capacity of COF-4 was significantly higher than that of all COFs reported to date, with a super high extraction efficiency of 6.84 mg/g/day. Further, the extraction efficiency of COF-4 was much higher than that of other state-of-art catalysts and adsorbents. To the best of our knowledge, this extraction efficiency of COF-4 is the highest reported thus far for any material. Our results show that significant improvements in photocatalytic performance can be achieved by tuning the local pore characteristics through linker modification in COFs. This motivated a deep investigation of the relationship between COF structure and photocatalytic activity.

## Photocatalytic mechanism studies

To obtain deeper insights into the photocatalytic mechanism, we carried out photoluminescence (PL) lifetime, transient absorption spectra (TAS) measurements, and density functional theory (DFT) calculations on the COFs to verify the electron excitation states,

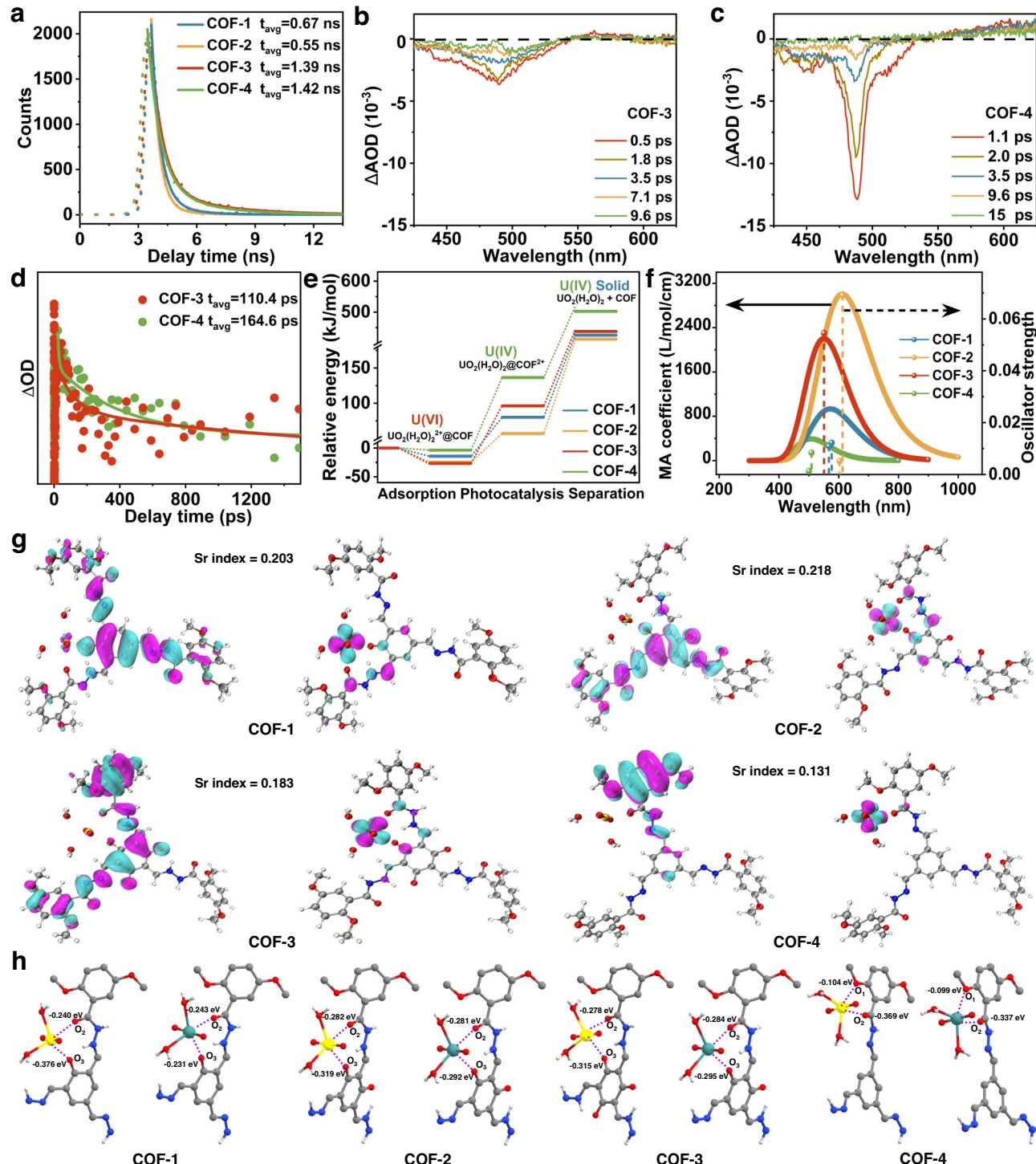

**Fig. 5 | Interpretation of photocatalytic mechanisms. a** PL lifetime of COF photocatalysts. **b, c** Time slices of the transient absorption spectra for COF-3 and COF-4, respectively. **d** Femtosecond time-resolved transient absorption decay kinetics of COF photocatalysts. **e** The relative free energy diagrams of uranium adsorption and reduction on COF photocatalysts. **f** The $S_1$ excited state oscillator strength and molar absorption coefficient for all four $[U(VI)O_2(H_2O_2)]@COF$ singlets. **g** The $S_1$ excited state electronic structures of COF photocatalysts (highlighting electron-hole distribution). **h** Hirshfeld charges of COFs before and after photocatalysis.

electronic donation sites, and transport pathways. Firstly, PL lifetime measurements were performed to estimate their excited-state lifetimes in the solid state. As shown in Fig. 5a, COF-1, COF-2, COF-3, and COF-4 showed PL lifetimes of 0.67, 0.55, 1.39, and 1.42 ns, respectively. The longer lifetimes for COF-3 and COF-4 indicated these particular COFs offered better interfacial charge separation and migration in

their extended conjugated skeletons, which was beneficial for improving their photocatalytic performance. Next, we collected TAS on COF-3 and COF-4 upon 365 nm pump pulsed laser excitation. Both samples showed strong bleaching bands at ~488 nm, indicating generation of the excited electrons (Fig. 5b, c). TAS kinetic plots with typical fitting curves of COF-3 and COF-4 determined the average

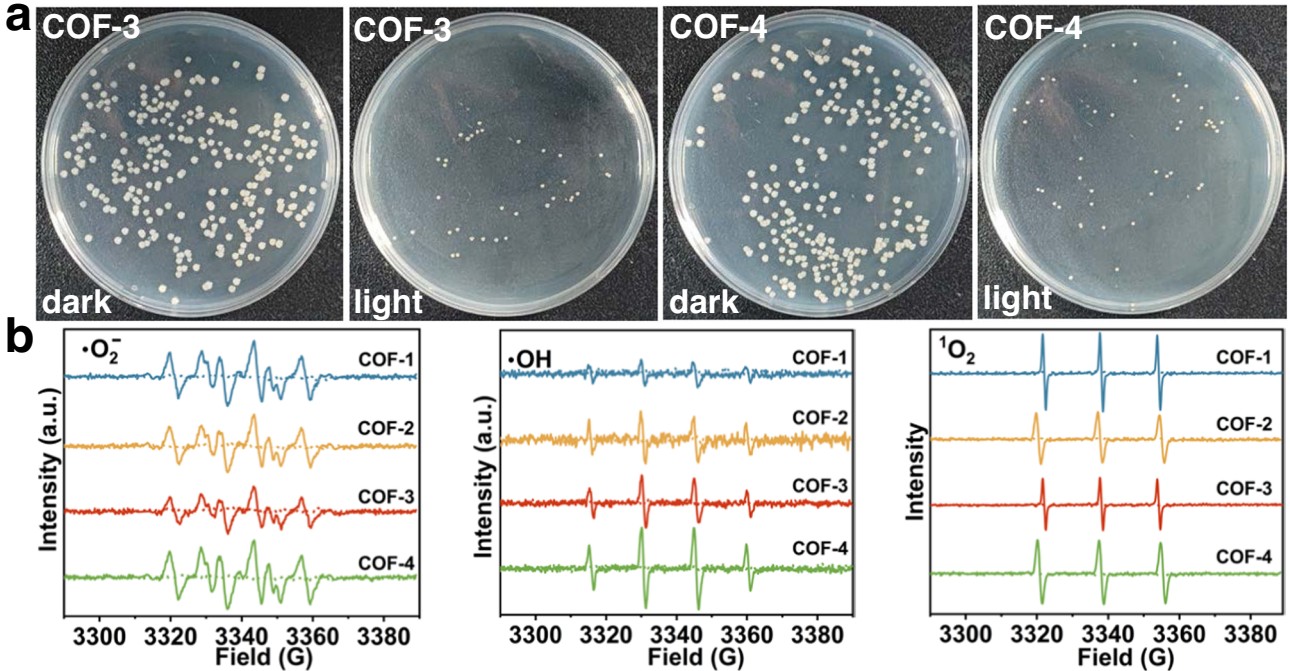

**Fig. 6 | Anti-biofouling activity and EPR measurements. a** Photographs of marine bacteria colonies after treatment with COF-3 and COF-4 in the dark and under visible light irradiation. **b** EPR spectra for •$O_2^-$-DMPO, •OH-DMPO, and $^1O_2$-TEMP complexes formed by visible light irradiation of COF-1, COF-2, COF-3, and COF-4.

lifetimes ($t_{avg}$) to be 110.4 and 164.6 ps, respectively (Fig. 5d), revealing COF-4 showed slower electron-hole combination kinetics (the resulting increased lifetime of charges is highly beneficial for photocatalysis).

Based on the above results and analyses, we next carried out DFT calculations to obtain deeper insights into the COFs at a molecular level. The relative free energy diagrams for uranium adsorption and reduction on COF-1, COF-2, COF-3, and COF-4 are shown in Fig. 5e. [U(VI)$O_2(H_2O_2)_2$]$^{2+}$ adsorbed spontaneously on all the COFs, with the strength of adsorption following the trend COF-3 > COF-2 > COF-1 > COF-4, which was consistent with the experimental results (Supplementary Fig. 10). This suggested that ketoenamine groups boosted U(VI) adsorption. The transfer of two electrons to uranium cations in [U(VI)$O_2(H_2O_2)_2$]$^{2+}$@COF under visible light irradiation yielded [U(IV)$O_2(H_2O)_2$]@COF$^{2+}$. The free energies of [U(IV)$O_2(H_2O)_2$]@COF-1$^{2+}$, [U(IV)$O_2(H_2O)_2$]@COF-2$^{2+}$, [U(IV)$O_2(H_2O)_2$]@COF-3$^{2+}$, and [U(IV)$O_2(H_2O)_2$]@COF-4$^{2+}$ were calculated to be 80.2, 57.2, 96.3, and 136.4 kJ/mol, respectively, suggesting that the visible light utilization ability followed the order COF-4 > COF-3 > COF-1 > COF-2[65,66]. Finally, [U(IV)$O_2(H_2O)_2$]@COF-4$^{2+}$ showed the highest free energy for the release of U(VI)$O_2(H_2O)_2$, explaining the superior photocatalytic performance of COF-4 for uranyl ion reduction to $UO_2$. Considering the competitive relationship between fluorescence emission from the $S_1$ excited state (from $S_1$ to ground state) and intersystem crossing (ISC) from the $S_1$ to $T_1$ state (Fig. 1a), we further calculated the $S_1$ excited state oscillator strength and molar absorption coefficient for all four [U(VI)$O_2(H_2O)_2$]$^{2+}$@COF singlet states. As expected, [U(VI)$O_2(H_2O)_2$]$^{2+}$@COF-4 shows a lower $S_1$ excited state oscillator strength and molar absorption coefficient strength compared to [U(VI)$O_2(H_2O)_2$]$^{2+}$@COF-1, [U(VI)$O_2(H_2O)_2$]$^{2+}$@COF-2, and [U(VI)$O_2(H_2O)_2$]$^{2+}$@COF-3, further explaining the superior photocatalytic activity of COF-4 (Fig. 5f).

The $S_1$ excited state electronic structures of the four COFs were further calculated at the PBE$_0$-D$_3$/SVP + SDD level using the time-dependent density functional theory (TD-DFT) method. Figure 5g shows the electronic distribution, electronic donation sites, and electron transport pathways from the COF donor to U(VI) acceptor. The

analysis clearly reveals that the DETH linker served as the electron transport site for COF-4, whilst asymmetric TFP or DTFP derivatives served as electron donor sites for COF-1 and COF-2, respectively. The electron transport site of COF-3 is the DETH together with symmetric ketoenamine moieties. The calculated integral of $Sr$ ($Sr$ index) for COF-1, COF-2, COF-3, and COF-4 were 0.203, 0.218, 0.183, and 0.134, respectively. A lower $Sr$ index indicates more efficient charge carrier separation. Moreover, the electron delocalization index results are as follows COF-4 (27.09) > COF-3 (26.88) > COF-2 (24.97) > COF-1 (24.5), which is consistent with the $Sr$ index results. Hirshfeld charges on the electron transfer sites of the COFs before and after electron transfer were obtained by the population analysis. The Hirshfeld charges dropped significantly at both the $O_1$ and $O_2$ sites of COF-4 following electron transfer to U(VI), confirming they served as the electron transfer sites during photocatalysis. In comparison, a decrease in the Hirshfeld charge was only observed at the $O_3$ sites in COF-1 and COF-2, with the $O_2$ sites remaining almost unchanged, indicating that the $O_3$ sites were the electron transfer sites during photocatalysis. The Hirshfeld charges at both $O_2$ and $O_3$ sites decreased in COF-3, suggesting that both linkers served as electron transfer sites (Fig. 5h)[67–69]. In addition, the bond lengths in the COF photocatalysts before and after electron transfer are summarized in Supplementary Table 7. During the photoreduction of U(VI) to U(IV), the $U_0$···$O_1$ and $U_0$···$O_2$ distances of COF-4 increased significantly from 2.80 to 3.245 and 2.30 to 2.522, respectively. Similarly, the $U_0$···$O_2$ and $U_0$···$O_3$ distances for COF-1 to COF-3 also increased after the photoreaction. All these results suggested that $O_1$ and $O_2$ sites from the DETH linker served as the electron transfer sites for COF-4. For COF-1 and COF-2, TFP or DTFP derivatives provided the electron transfer platform. Both DETH and TP ketoenamine moieties are active for electron transfer in COF-3. Accordingly, the structure of the COFs controlled the mechanism of electronic transfer during photocatalytic U(VI) reduction.

## Anti-biofouling studies
The presence of marine microorganisms in seawater hampers the performance of adsorbents for uranium capture through marine

biofouling. Accordingly, we carried out anti-biofouling studies on COF-3 and COF-4. As shown in Fig. 6 and Supplementary Table 8, the inhibition rates of marine bacteria were 85.88% and 89.71% for COF-3 and COF-4, respectively, under visible light irradiation, which were vastly superior to the inhibition rates under dark conditions. Next, we carried out EPR experiments to identify the active species responsible for the anti-biofouling activity. No radical signals were detected under dark conditions (Fig. 6b). As expected, the $\cdot O_2^-/\cdot OH$ and $^1O_2$-were trapped by 3,4-dihydro-2,3-dimethyl-2H-pyrrole 1-oxide (DMPO) and 2,2,6,6-tetramethylpiperidine (TEMP) trapping agents, respectively, under visible light irradiation. These results reveal that $\cdot O_2^-$, $\cdot OH$, and $^1O_2$ species were generated by the COF photocatalysts under visible light irradiation. These reactive oxygen species impart the COFs with potent anti-biofouling properties against marine bacteria in seawater.

## Discussion

The aforementioned experimental and theoretical findings demonstrate that the developed COFs exhibited promise as photocatalysts for uranium extraction from seawater and groundwater. By varying the electronic and local pore characteristics of the COFs through linker modification, the factors influencing the photocatalytic properties of the COFs could be understood. COF-4 demonstrated the best photocatalytic performance, which could be attributed to the $S_1$ excited state electronic distribution and efficient electron transfer to a triplet state under visible light, enabling efficient charge carrier utilization with minimal energy loss during photocatalytic reduction of U(VI). On the basis of the excellent photocatalytic performance of COF-4, we estimated the cost of synthesizing COF-4 to be ~$2.7 USD/g, and the cost for uranium extraction to be ~$4.3 USD/g (based on reuse of the photocatalyst and its decay of activity), suggesting economic feasibility for practical uranium extraction from seawater. Results guide the development of high-performance COF photocatalysts for uranium extraction from seawater, wastewater, and contaminated groundwater.

In summary, we report a new design strategy for constructing highly conjugated hydrazide-based COF photocatalysts with unique optoelectronic characteristics and outstanding photocatalytic activities. By optimizing the excited state electron distribution, electronic donation sites, and electron transportation in the COFs, photocatalytic U(VI) extraction from seawater and wastewater could be maximized. One of our COFs, COF-4, possessed an extremely high uranium uptake capacity of 6.84 mg/g/day, state-of-the-art performance for any COF-based adsorbent or adsorbent-photocatalyst in natural seawater. The new mechanistic insights this work provides about charge separation and transport in COFs during photocatalysts support the rational design of new COFs for uranium extraction and other photocatalytic applications.

## Methods

### Materials and measurements

All chemicals were sourced from commercial suppliers and used without further purification. The seawater was collected in Maoming, Guangdong Province, China. The groundwater was collected in Mentougou, Beijing, China. Powder X-ray diffraction (PXRD) patterns were collected on a Rigaku SmartLab SE X-ray diffractometer equipped with a Cu Kα source (small angle X-ray scattering data collected on a Bruker D8 Advance diffractometer were used to correct the deviation). BET surface areas were determined from $N_2$ adsorption/desorption isotherms collected at 77 K using a Micromeritics TriStar II. Scanning electron microscopy (SEM) images were recorded on a Hitachi SU 8100 Scanning Electron Microscope. Fourier transform infrared spectra (FT-IR) were recorded on a SHIMADZU IRTracer-100. Transmission electron microscopy (TEM) and high-resolution transmission electron microscopy (HRTEM) images were recorded on a JEOL JEM-2100F transmission electron microscope operating at an accelerating

voltage of 200 kV. Solid-state $^{13}C$ CP/MAS NMR spectra were collected on a Bruker AVANCE III 400 WB spectrometer. X-ray photoelectron spectroscopy (XPS) analyses were performed using a Thermo Scientific ESCALAB 250Xi spectrometer, equipped with a monochromatic Al $K_\alpha$ X-ray source. Photoelectrochemical experiments measurements were performed on a CHI760 workstation. Electron paramagnetic resonance (EPR) spectra were obtained on a Bruker A200 spectrometer. UV-vis spectroscopy results were recorded in diffuse reflectance (DR) mode at room temperature on a SHIMADZU UV-2700 spectrophotometer equipped with an integrating sphere attachment. Thermogravimetric analyses (TGA) were carried out on a NETZSCH STA 2500 instrument. Inductively coupled plasma mass spectrometry (ICP-MS) analyses were performed on an Agilent 7800 spectrometer. Photoluminescence (PL) lifetime data were measured on an Edinburgh Instruments FLS1000 spectrometer. Femtosecond Transient absorption spectra (TAS) measurements were carried out on an Ultrafast systems HELIOS spectrometer. U $L_{III}$-edge X-ray absorption spectroscopy (XAS) data were collected in transmission mode at the Shanghai Synchrotron Radiation Facility (14 W station, SSRF).

### Synthesis of COF-1

In a typical synthesis, 2-hydroxybenzene-1,3,5-tricarbaldehyde (TFP, 14.3 mg) and 2,5-diethoxy terephthalo hydrazide (DETH, 33.9 mg) were dissolved in a mixed solvent solution containing mesitylene (0.32 mL)/1,4-dioxane (0.48 mL)/acetic acid (6 M, 0.08 mL) in a 5 mL glass tube. Next, the mixture was sonicated and frozen in a liquid nitrogen bath and sealed with a gas torch. The tube was then heated at 120 °C for 72 h, after which the product was collected by filtration, and washed several times with ethanol, yielding COF-1.

### Synthesis of COF-2

In a typical synthesis, 2,4-dihydroxybenzene-1,3,5-tricarbaldehyde (DTFP, 15.5 mg) and DETH (33.9 mg) were dissolved in a mixed solvent solution containing mesitylene (0.32 mL)/1,4-dioxane (0.48 mL)/acetic acid (6 M, 0.08 mL) in a 5 mL glass tube. Next, the mixture was sonicated and frozen in a liquid nitrogen bath and sealed with a gas torch. The tube was then heated at 120 °C for 72 h, after which the product was collected by filtration, and washed several times with ethanol, yielding COF-2.

### Synthesis of COF-3

In a typical synthesis, 1,3,5-triformylphloroglucinol (TP, 16.8 mg) and DETH (33.9 mg) were dissolved in a mixed solvent solution containing mesitylene (0.32 mL)/1,4-dioxane (0.48 mL)/acetic acid (6 M, 0.08 mL) in a 5 mL glass tube. Next, the mixture was sonicated and frozen in a liquid nitrogen bath and sealed with a gas torch. The tube was then heated at 120 °C for 72 h, after which the product was collected by filtration, and washed several times with ethanol, yielding COF-3.

### Synthesis of COF-4

In a typical synthesis, 1,3,5 triformylbenzene (TFB, 13 mg) and DETH (33.9 mg) were dissolved in a mixed solvent solution containing mesitylene (0.32 mL)/1,4-dioxane (0.48 mL)/acetic acid (6 M, 0.08 mL) in a 5 mL glass tube. Next, the mixture was sonicated and frozen in a liquid nitrogen bath and sealed with a gas torch. The tube was then heated at 120 °C for 72 h, after which the product was collected by filtration, and washed several times with ethanol, yielding COF-4.

### Photocatalytic uranium extraction studies

The performance of the COFs for the photocatalytic reduction of U(VI) were evaluated in a photoreactor under visible light irradiation from a 300 W xenon lamp (PerfectLight, PLS-SXE300D). 10 mg of the photocatalyst (COF-1, COF-2, COF-3, or COF-4) was dispersed in 100 mL of a uranyl spiked solution (~20 ppm in seawater or ~10 ppm in

groundwater). The solutions were stirred overnight in the dark at 25 °C to achieve adsorption equilibrium. Subsequently, the reactor was continuously exposed to the Xe lamp. At regular intervals, aliquots of the dispersion were removed and filtered through a 0.45 μm membrane filter. The concentration of U(VI) in the filtrates was measured by UV-vis spectrophotometry at a wavelength of 650 nm using the Arsenazo III method. After the photocatalytic experiments, COF-4 was washed with a $HNO_3$ (pH = 3)/$NaNO_3$ solution and distilled water several times. After filtration, COF-4 was collected for subsequent reuse. This process involves trace COF loss (determined gravimetrically). Further photocatalytic experiments were conducted using natural seawater. The seawater used in the work was filtered to remove any insoluble particulates. 9 mg of COF-4 was spread on the top of a column filled with sea sand, then the seawater cycled in a continuous loop through the column from top to bottom. The Xe lamp irradiated the COF-4 with visible light irradiation from above. The filtrate at the bottom of the sand column was analyzed periodically using ICP-MS to quantify the remaining uranium content.

## Data availability
The authors declare that all the data supporting the findings of this study are available within the article (and Supplementary Information Files), or available from the corresponding author on request. Source data are provided with this paper.

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

## Acknowledgements

We gratefully acknowledge funding support from the National Key Research and Development Program of China (2018YFC1900105), National Science Foundation of China (Grants U2167218; 22006036; 22276054), the Beijing Outstanding Young Scientist Program (H.Y., Z.C, and X.W.), and the Robert A. Welch Foundation (B – 0027) (S.M.). We also acknowledge support from the 14 W station in Shanghai Synchrotron Radiation Facility (SSRF). G.I.N.W. acknowledges funding support from the Royal Society Te Apārangi (for the award of a James Cook Research Fellowship).

## Author contributions

H.Y., X.W., and S.M. conceived and designed the research. Z.C., J.W., and M.H. performed the synthesis and characterization. Z.C. and Y.X. carried out the photocatalysis tests. Z.C. and M.H. contributed to DFT calculation. Y.X. and X.L. performed and analyzed the EXAFS. H.Y., X.W., G.I.N.W., and S.M. wrote the manuscript. All authors contributed to the discussion, and gave approval to the final version of the manuscript.

## Competing interests

The authors declare no competing interests.
