## [Peer Review File · Nature Communications]

Tuning Excited State Electronic Structure and Charge Transport in Covalent Organic Frameworks for Enhanced Photocatalytic PerformanceReviewers' Comments:

Reviewer #1:

Remarks to the Author:

In this manuscript, Chen et al report a series of isorecticular COFs with different excited state electron distributions, charge transport properties, and local pore characteristics, which allowed exploration of the relationship between structural features and photocatalytic activity. It is interesting that COF-4 in the absence of carbonyl groups exhibited a remarkably excited state electron utilization efficiency and charge transfer properties, resulting in an excellent photocatalytic uranium extraction performance of ~6.84 mg/g/day in natural seawater. The uranium extraction efficiency of COF-4 is comparable to the best materials reported thus far. This is very interesting and promising research work. The manuscript is well organized and shows high novelty and a broad range of interests. Based on these mentions, I recommend the acceptance of this manuscript after a minor revision.

1. The authors claimed all COFs were calculated to be AA stacking simulations. However, the authors didn't compare the PXRD with AB stacking. The AB stacking simulation should be provided for comparison.

2. How was the cycling experiment performed? The authors should describe this in more detail in the experimental section.

3. The authors showed COF-4 is a promising photocatalyst for potential uranium extraction applications. How is the cost of uranium extraction from seawater?

4. Two related references should be cited. Please see "Synthesis of Vinylene-Linked Covalent Organic Frameworks by Monomer Self-Catalyzed Activation of Knoevenagel Condensation. *J. Am. Chem. Soc.* 2022, 144, 3653. Doi: 10.1021/jacs.1c12902" "Two-dimensional semiconducting covalent organic frameworks via condensation at arylmethyl carbon atoms. *Nat. Commun.* 2019, 10, 2467. Doi: 10.1038/s41467-019-10504-6"

5. Some of the data in Supplementary Table 6 was not presented in Fig.4. I suggest summarizing all the data in Fig.4 as well.

Reviewer #2:

Remarks to the Author:

This manuscript reported a simple strategy for producing a family of isorecticular crystalline hydrazide-based COF photocatalysts. The study of photocatalytic uranium extraction performances indicates high extraction efficiency towards U(VI) in seawater and groundwater. Significantly, the uranium extraction capacity reached a record number of ~6.84 mg/g/day among all COF materials in seawater. These results delineate important synthetic advances toward the implementation of COFs in uranium extraction in seawater and environmental remediation. DFT calculations revealed that the photocatalytic performance could be attributed to the S1 excited state electronic distribution and efficient electron transfer to a triplet state under visible light, enabling efficient charge carrier utilization with minimal energy loss during photocatalytic reduction of U(VI). Overall, this work is a highly important work representing a significant progress in the area of uranium extraction from seawater, which is supported with solid data and logical discussion. I therefore recommend it publishing in nature communications after addressing the following minor issues.

1. It is recommended to bring some important information from supplementary information to the manuscript, such as methods of photocatalytic uranium extraction.

2. COF-4 showed good selectivity for uranium in natural seawater. What is the difference or benefit of photocatalysis compared to the physiochemical sorption method? Amidoxime functional sorbents exhibit excellent selectivity to uranium than other metals.

3. The authors state that "...[U(IV)O₂(H₂O)₂]@COF-4²⁺ showed the highest free energy for the release of U(VI)O₂(H₂O)₂, explaining the superior photocatalytic performance of COF-4 for uranyl ion reduction to UO₂." Where or which part offered the free energy? The authors should make an explanation.

4. In seawater, uranium is in the speciation of [UO₂(CO₃)₃]⁴⁻. Therefore, some discussion needs to

be modified.

5. Bulk PXRD revealed the stability of COF-4 after photocatalysis. How about the BET surface area?

6. Uranium photocatalytic reduction without sacrificial reagent is very impressive. I am curious what species is oxidized during the uranium photoreduction process. Please add some discussion to clarify this issue.

Reviewer #3:

Remarks to the Author:

The authors described in this manuscript the synthesis of a series of isorecticular covalent organic frameworks with different electronic properties and local pore characteristics together with the investigation of their performances in the photocatalytic extraction of uranium from seawater. The authors illustrated that the light-harvesting capability, electronic band gap, excited state electronic distribution and transportation of COF photocatalysts can be finely tuned via systematically modulating the local pore characteristics. The synthesized COFs were investigated in detail by a variety of characterization techniques to correlate with their photocatalytic activities. The elucidated structural information in combination with a host of experimental methods and theoretical calculations facilitates the production of a clear map for the excited state electronic distribution and transport pathways in the COFs. Uranium in the form of UO₂ nanoparticles was identified after the photocatalysis, showing the potential of synthesized COFs for uranium extraction under various conditions. COF-3 and COF-4 further demonstrated good anti-biofouling performance. This study offers a new insight for understanding the photocatalysis mechanism of COFs by varying their excited state electronic distribution and electron transport properties from donor to acceptor. Overall speaking, this work was well-conceived with adequate comprehension and in-depth and will represent a significant contribution to the community. This reviewer recommends its acceptance for publication in Nature Communications after minor revisions with the following comments addressed.

(1) The U(VI) was reduced to U(IV) through photocatalysis. If it is possible to confirm this by PXRD, it would help to include index lines for all UO₂ reflections in the PXRD.

(2) Authors should provide a TEM image for the ultimate product after photocatalysis in groundwater, so that there is sufficient evidence to confirm the final product.

(3) Details on how the recyclability of COF 4 for photocatalytic uranium extraction is performed should be supplied. How the UO₂ was collected?

(4) Why did COFs show anti-biofouling activities in seawater?

Point-by-Point Response to Reviewers' Comments

We greatly appreciate the very positive comments and constructive suggestions from all reviewers and editorial office, and we have revised the manuscript accordingly, as detailed in the responses below.

REVIEWER COMMENTS

Reviewer #1 (Remarks to the Author):

In this manuscript, Chen et al report a series of iso-reticular COFs with different excited state electron distributions, charge transport properties, and local pore characteristics, which allowed exploration of the relationship between structural features and photocatalytic activity. It is interesting that COF-4 in the absence of carbonyl groups exhibited a remarkably excited state electron utilization efficiency and charge transfer properties, resulting in an excellent photocatalytic uranium extraction performance of ~6.84 mg/g/day in natural seawater. The uranium extraction efficiency of COF-4 is comparable to the best materials reported thus far. This is very interesting and promising research work. The manuscript is well organized and shows high novelty and a broad range of interests. Based on these mentions, I recommend the acceptance of this manuscript after a minor revision.

Response: We are grateful to the reviewer for taking the time to evaluate our work and greatly appreciate the positive comments and support of our work.

Comment 1: The authors claimed all COFs were calculated to be AA stacking simulations. However, the authors didn't compare the PXRD with AB stacking. The AB stacking simulation should be provided for comparison.

Response: We have amended the manuscript to incorporate this excellent suggestion. The AB stacking simulations have been added to the PXRD patterns for comparison in the revised manuscript. The comparison further supports our initial conclusion that all COFs possessed AA stacking.

Comment 2: How was the cycling experiment performed? The authors should describe this in more detail in the experimental section.

Response: We have amended the manuscript to incorporate this excellent suggestion. After photocatalytic experiments, COF-4 was washed with a HNO₃ (pH=3)/NaNO₃ solution and distilled water. After filtration, COF-4 was collected for subsequent reuse. This process involves trace COF loss (determined gravimetrically). The generated UO₂ could be removed from COF-4 by sonication, centrifugation, and filtration after each test or the recycling tests. A small amount of the UO₂ remained attached to the COF after these processes.

Comment 3: The authors showed COF-4 is a promising photocatalyst for potential uranium extraction applications. How is the cost of uranium extraction from seawater?

Response: We have amended the manuscript to incorporate this excellent suggestion. We estimated the cost of synthesizing COF-4 to be ~\$2.7 USD/g, suggesting the economic feasibility of the adsorbent-photocatalyst. Moreover, the cost for uranium extraction using COF-4 was estimated to be ~\$4.3 USD per gram (based on reuse of the photocatalyst and its decay of activity).

Comment 4: Two related references should be cited. Please see "Synthesis of Vinylene-Linked Covalent Organic Frameworks by Monomer Self-Catalyzed Activation of Knoevenagel Condensation. J. Am. Chem. Soc. 2022, 144, 3653. Doi: 10.1021/jacs.1c12902" "Two-dimensional semiconducting

covalent organic frameworks via condensation at arylmethyl carbon atoms. Nat. Commun. 2019, 10, 2467. Doi: 10.1038/s41467-019-10504-6”

Response: We appreciate the comment from the reviewer. The above references have now been cited in the revised manuscript.

Comment 5: Some of the data in Supplementary Table 6 was not presented in Fig.4. I suggest summarizing all the data in Fig.4 as well.

Response: We appreciate the reviewer’s constructive comments. The data described in Supplementary Table 6 is now also compared in Fig. 4.

Reviewer #2 (Remarks to the Author):

This manuscript reported a simple strategy for producing a family of isorecticular crystalline hydrazide-based COF photocatalysts. The study of photocatalytic uranium extraction performances indicates high extraction efficiency towards U(VI) in seawater and groundwater. Significantly, the uranium extraction capacity reached a record number of ~6.84 mg/g/day among all COF materials in seawater. These results delineate important synthetic advances toward the implementation of COFs in uranium extraction in seawater and environmental remediation. DFT calculations revealed that the photocatalytic performance could be attributed to the S1 excited state electronic distribution and efficient electron transfer to a triplet state under visible light, enabling efficient charge carrier utilization with minimal energy loss during photocatalytic reduction of U(VI). Overall, this work is a highly important work representing a significant progress in the area of uranium extraction from seawater, which is supported with solid data and logical discussion. I therefore recommend it publishing in nature communications after addressing the following minor issues.

Response: We are grateful to the reviewer for taking the time to evaluate our work and greatly appreciate the positive comments and support of our work.

Comment 1: It is recommended to bring some important information from supplementary information to the manuscript, such as methods of photocatalytic uranium extraction.

Response: We appreciate the constructive comment from the reviewer. We have moved the methods of photocatalytic uranium extraction section from the Supplementary Information to the revised manuscript.

Comment 2: COF-4 showed good selectivity for uranium in natural seawater. What is the difference or benefit of photocatalysis compared to the physiochemical sorption method? Amidoxime functional sorbents exhibit excellent selectivity to uranium than other metals.

Response: We appreciate the constructive comment from the reviewer. The coexistence of a variety of other metal ions in seawater, together with marine biofouling which can passivate adsorbents, make selective uranium extraction from seawater challenging. Amidoxime-functionalized adsorbents have been widely used to improve the adsorption selectivity towards uranyl ions due to the strong interaction between amidoxime groups and uranyl (UO_2^{2+}), thereby delivering both high adsorption capacities and fast kinetics. However, many adsorbents containing amidoxime groups also have modest affinities for vanadium and copper ions in natural seawater, reducing the efficiency of uranium extraction. The co-adsorption of U, V, and Cu ions is undesirable, as the separation of these ions increases the overall cost of U extraction from seawater. In our work, the developed COF-4 photocatalyst showed a high selectivity towards uranium over other metal ions, particularly copper and vanadium, suggesting good potential for real-world uranium extraction from seawater.

Comment 3: The authors state that "...[U(IV)O₂(H₂O)₂]@COF-4²⁺ showed the highest free energy for the release of U(VI)O₂(H₂O)₂, explaining the superior photocatalytic performance of COF-4 for uranyl ion reduction to UO₂." Where or which part offered the free energy? The authors should make an explanation.

Response: We appreciate the comment from the reviewer. [U(VI)O₂(H₂O)₂]²⁺ adsorbed spontaneously on all the COFs, with the strength of adsorption following the trend COF-3 > COF-2 > COF-1 > COF-4, which was consistent with the experimental results. Subsequently, the transfer of two electrons from the COF to uranium cations in [U(VI)O₂(H₂O)₂]²⁺@COF during the photocatalytic reduction processes under visible light irradiation yielded [U(IV)O₂(H₂O)₂]@COF²⁺ (the photocatalytic ability following the trend COF-4 > COF-3 > COF-1 > COF-2). Afterward, the attached U(IV)O₂(H₂O)₂ detached from the COFs with the system free energy following COF-4 > COF-3 > COF-1 > COF-2, with this order reflecting the relative photocatalytic activity of the COFs. The DFT calculation results show that COF-4 exhibited the highest free energy for the release of U(VI)O₂(H₂O)₂, suggesting the product (U(VI)O₂(H₂O)₂) strongly binds to the photocatalyst.

Comment 4: In seawater, uranium is in the speciation of [UO₂(CO₃)₃]⁴⁻. Therefore, some discussion needs to be modified.

Response: We appreciate the reviewer's constructive comments. The uranium exists mainly as [UO₂(CO₃)₃]⁴⁻ in natural seawater, and the adsorbent needs to compete with the carbonate group for the uranyl ions. In comparison with the binding affinity of CO₃²⁻ to UO₂²⁺, hydrazine-carbonyl groups through COFs 1 to 3 and the hydrazine sites in COF-4 showed stronger binding affinities towards UO₂²⁺ than towards CO₃²⁻, therefore the [U(VI)O₂(H₂O)₂]²⁺ adsorbed spontaneously on all the COFs (*Cell Rep. Phys. Sci.* 2023, 4, 101220). The competition of the adsorbent and carbonate for uranyl has been reported in other works, such as *Nat. Sustain.* 2021, 4, 708-714; *Adv. Mater.* 2018, 30, 1705479; *J. Am. Chem. Soc.* 2021, 143, 14523-14529; *Chem* 2022, 8, 2749-2765; *Energy Environ. Sci.* 2022, 15, 3462-3469. We have added the discussion in the revised manuscript.

Comment 5: Bulk PXRD revealed the stability of COF-4 after photocatalysis. How about the BET surface area?

Response: We appreciate the reviewer's constructive comments. We carried out the N₂ adsorption-desorption measurement on COF-4 after photocatalysis, revealing that the porosity was retained. Moreover, the BET surface area of the used COF-4 was calculated to be 547.4 m²/g, revealing excellent stability of COF-4.

Comment 6: Uranium photocatalytic reduction without sacrificial reagent is very impressive. I am curious what species is oxidized during the uranium photoreduction process. Please add some discussion to clarify this issue.

Response: We appreciate the comment from the reviewer. The photogenerated electrons reduced U(VI) to U(IV), whilst photogenerated holes were consumed via the generation of reactive oxygen species (e.g., •OH) or oxidation of organic species in seawater under visible light irradiation.

Reviewer #3 (Remarks to the Author):

The authors described in this manuscript the synthesis of a series of isorecticular covalent organic frameworks with different electronic properties and local pore characteristics together with the investigation of their performances in the photocatalytic extraction of uranium from seawater. The

authors illustrated that the light-harvesting capability, electronic band gap, excited state electronic distribution and transportation of COF photocatalysts can be finely tuned via systematically modulating the local pore characteristics. The synthesized COFs were investigated in detail by a variety of characterization techniques to correlate with their photocatalytic activities. The elucidated structural information in combination with a host of experimental methods and theoretical calculations facilitates the production of a clear map for the excited state electronic distribution and transport pathways in the COFs. Uranium in the form of UO₂ nanoparticles was identified after the photocatalysis, showing the potential of synthesized COFs for uranium extraction under various conditions. COF-3 and COF-4 further demonstrated good anti-biofouling performance. This study offers a new insight for understanding the photocatalysis mechanism of COFs by varying their excited state electronic distribution and electron transport properties from donor to acceptor. Overall speaking, this work was well-conceived with adequate comprehension and in-depth and will represent a significant contribution to the community. This reviewer recommends its acceptance for publication in Nature Communications after minor revisions with the following comments addressed.

Response: We are grateful to the reviewer for taking the time to evaluate our work and greatly appreciate the positive comments and support of our work.

Comment 1: The U(VI) was reduced to U(IV) through photocatalysis. If it is possible to confirm this by PXRD, it would help to include index lines for all UO₂ reflections in the PXRD.

Response: We appreciate the reviewer's comments. We carried out PXRD measurement on COF-4 after the photocatalysis. However, no UO₂ reflections were observed in the PXRD pattern. This is due to the low concentration of UO₂ products and the small size of the UO₂ nanoparticles. Nevertheless, XPS, XAS, HRTEM results characterizations studies identified UO₂ nanoparticles as the solid product.

Comment 2: Authors should provide a TEM image for the ultimate product after photocatalysis in groundwater, so that there is sufficient evidence to confirm the final product.

Response: We appreciate the constructive comment from the reviewer. We carried out HRTEM measurements on COF-4 after the photocatalysis in uranium-spiked groundwater. The results revealed the formation of UO₂ after photocatalysis.

Comment 3: Details on how the recyclability of COF 4 for photocatalytic uranium extraction is performed should be supplied. How the UO₂ was collected?

Response: We have amended the manuscript to incorporate this excellent suggestion. After photocatalytic experiments, COF-4 was washed with a HNO₃ (pH=3)/NaNO₃ solution and distilled water. After filtration, COF-4 was collected for subsequent reuse. This process involves trace COF loss (determined gravimetrically). The generated UO₂ could be removed from COF-4 by sonication, centrifugation, and filtration after each test or the recycling tests. A small amount of the UO₂ remained attached to the COF after these processes.

Comment 4: Why did COFs show anti-biofouling activities in seawater?

Response: We appreciate the constructive comment from the reviewer. The photogenerated superoxide radicals ($\cdot\text{O}_2^-$), singlet oxygen ($^1\text{O}_2$), and hydroxyl radicals ($\cdot\text{OH}$) formed under visible light irradiation of the COF can damage the cell walls of Marine microorganisms. The anti-biofouling experiments and EPR studies confirmed the anti-biofouling activities of developed COFs.

Again, we thank all reviewers for the constructive suggestions, which have made our manuscript much improved.

Sincerely,

Shengqian Ma, PhD

Professor and Welch Chair in Chemistry

Reviewers' Comments:

Reviewer #1:

Remarks to the Author:

In this revised version, the authors have well-resolved the concerns from the reviewers, thus I recommend accepting this work for publication at its current state.

Reviewer #2:

Remarks to the Author:

Authors have fully addressed my comments during the revision. This work can be published in its current form.

Reviewer #3:

Remarks to the Author:

Authors addressed all comments properly. I suggest the publication as is.

Point-by-Point Response to Reviewers' Comments

We greatly appreciate the very positive comments and support of our work from all reviewers and editorial office, and we have revised the manuscript accordingly, as detailed in the responses below.

Reviewer #1 (Remarks to the Author):

In this revised version, the authors have well-resolved the concerns from the reviewers, thus I recommend accepting this work for publication at its current state.

Response: We are grateful to the reviewer for taking the time to evaluate our work and greatly appreciate the positive comments and support of our work.

Reviewer #2 (Remarks to the Author):

Authors have fully addressed my comments during the revision. This work can be published in its current form.

Response: We are grateful to the reviewer for taking the time to evaluate our work and greatly appreciate the positive comments and support of our work.

Reviewer #3 (Remarks to the Author):

Authors addressed all comments properly. I suggest the publication as is.

Response: We are grateful to the reviewer for taking the time to evaluate our work and greatly appreciate the positive comments and support of our work.